



# A new gas absorption optical depth parameterisation for RTTOV v13

James Hocking[1], Jérôme Vidot[2], Pascal Brunel[2], Pascale Roquet[2], Bruna Silveira[2], Emma Turner[1], and Cristina Lupu[3]

[1]Met Office, Fitzroy Road, Exeter, EX1 2PB, UK
[2]CNRM, Université de Toulouse, Météo-France, CNRS, Lannion, France
[3]ECMWF, Shinfield Park, Reading, UK

**Correspondence:** James Hocking (james.hocking@metoffice.gov.uk)

**Abstract.** This paper describes a new gas optical depth parameterisation implemented in the most recent release, version 13, of the radiative transfer model RTTOV (Radiative Transfer for TOVS). RTTOV is a fast, one-dimensional radiative transfer model for simulating top-of-atmosphere visible, infrared and microwave radiances observed by downward-viewing space-borne passive sensors. A key component of the model is the fast parameterisation of absorption by the various gases in the

5 atmosphere. The existing parameterisation in RTTOV has been extended over many years to allow for additional variable gases in RTTOV simulations and to account for solar radiation and better support geostationary sensors by extending the validity to higher zenith angles. However, there are limitations inherent in the current approach which make it difficult to develop it further, for example by adding new variable gases. We describe a new parameterisation that can be applied across the whole spectrum, allows for a wide range of zenith angles in support of solar radiation and geostationary sensors, and for which it will be easier

10 to add new variable gases in support of user requirements. Comparisons against line-by-line radiative transfer simulations, and against observations in the ECMWF operational system yield promising results, suggesting that the new parameterisation generally compares well with the old one in terms of accuracy. Further validation is planned, including testing in operational numerical weather prediction data assimilation systems.

## 1 Introduction

RTTOV (Radiative Transfer for TOVS) (Saunders et al., 2018) is a fast, one-dimensional radiative transfer model for simulating

20 top-of-atmosphere visible, infrared (IR) and microwave radiances observed by downward-viewing space-borne passive sensors. RTTOV was originally developed at the European Centre for Medium-Range Weather Forecasts (ECMWF) in the 1990s





to enable the direct assimilation of radiances in their operational numerical weather prediction system and since 1998 has undergone much development within the EUMETSAT-funded Numerical Weather Prediction Satellite Applications Facility (NWP SAF). Today RTTOV is widely used around the world in a range of applications including operational data assimilation in NWP (Lupu and Geer, 2015), physical retrievals using satellite data (Ghent et al., 2017), generating simulated satellite imagery from NWP models (Lupu and Wilhelmsson, 2016), and studies assessing future satellite instruments (Andrey-Andrés et al., 2018). In order to support variational assimilation and retrieval applications, RTTOV comprises not only a direct (or forward) model, but also tangent linear, adjoint, and full Jacobian models.

A variety of fast radiative transfer models exist which employ different parameterisations to achieve the computational efficiency necessary for operational use. For example, the Optimal Spectral Sampling (OSS) model (Moncet et al., 2015) approximates channel transmittances as a weighted sum of monochromatic transmittances obtained from look-up tables at an optimised set of wavenumbers falling within the channel spectral response. Models based on Principal Components (PC) methods such as the Havemann Taylor Fast Radiative Transfer Code (HT-FRTC) (Havemann et al., 2018) and PC-RTTOV (Matricardi, 2010) predict PC scores for transmittance or radiance spectra. HT-FRTC uses monochromatic calculations at an optimised set of wavenumbers as the predictors for the PC scores while PC-RTTOV uses standard RTTOV simulated radiances for several hundred channels. Such models can be very efficient when computing full radiance spectra for hyperspectral sensors with many channels. The Community Radiative Transfer Model (CRTM) (Chen et al., 2008; Ding et al., 2011) follows a method with some similarities to that in RTTOV: layer optical depths are predicted by a linear regression onto variables computed from the input atmospheric profile. CRTM has a pool of such "predictors" from which an optimal selection is made for each optical depth being predicted. In contrast, RTTOV has a fixed set of predictors that are used for all layers for a given channel. RTTOV predicts optical depths on a set of layers defined by a fixed set of pressure levels. CRTM provides this option, but also allows instead for the parameterisation to operate on layers defined by fixed absorber concentration amounts.

The existing parameterisation of gas absorption optical depths employed by RTTOV has proved successful for many years in various operational and research applications, but poses challenges in respect of future developments of the model. This paper presents a new optical depth parameterisation that has been implemented in the recent major release of RTTOV, version 13. Section 2 gives an overview of the existing optical depth prediction scheme and discusses its weaknesses. Section 3 describes the new parameterisation. Section 4 briefly discusses the treatment of clear-sky Rayleigh scattering for visible channels. Section 5 presents validation results for the new parameterisation. A summary is given in Section 6.

## 2 The existing optical depth parameterisation

The pre-existing optical depth parameterisation implemented in RTTOV is based on the methods in McMillin and Fleming (1976) and Eyre and Woolf (1988) and is described in Saunders et al. (2018). The aim of the parameterisation is to obtain layer optical depths for a given satellite sensor channel for specific gases. The individual gas layer optical depths are summed to give the total channel layer optical depth due to gas absorption which is subsequently used in solving the radiative transfer equation to obtain top-of-atmosphere radiances. The gas layer optical depths are related to atmospheric "predictors" via a





linear regression. The predictors are quantities derived from the input atmospheric profile variables: pressure, temperature, gas concentrations and local zenith angle of the radiation path. The regression coefficients are obtained using layer optical depths obtained from channel-integrated transmittance profiles for a diverse set of training profiles using an accurate line-by-line (LBL) radiative transfer model. The training profiles are interpolated onto a fixed set of pressure levels giving a pre-defined set of layers for the optical depth regression.

The pre-existing parameterisation is trained using 83 profiles (Matricardi, 2008; Saunders et al., 2017) interpolated onto a fixed set of 54 pressure levels between 0.005 hPa and 1050 hPa defining 53 layers. In addition, optical depth coefficients are calculated on a set of 101 levels between 0.005 hPa and 1100 hPa (100 layers) for hyperspectral IR sounders. The LBLRTM model (Clough et al., 2005; Rothman et al., 2013) is used to generate level-to-space transmittance profiles at high spectral resolution in the visible and IR. The LBL transmittances are computed at a spectral resolution of 0.01 cm$^{-1}$ in the visible/near-IR

(2000–25500 cm$^{-1}$) and at a resolution of 0.001 cm$^{-1}$ in the IR (75–3325 cm$^{-1}$). For non-hyperspectral sensors, these high resolution transmittances are averaged at 1.0 cm$^{-1}$ (visible/near-IR) or 0.1 cm$^{-1}$ (IR) before being integrated over the channel spectral response functions in order to calculate the regression coefficients. For hyperspectral IR sounders, the LBL transmittances at 0.001 cm$^{-1}$ are integrated over the channel spectral response functions directly. At the time of writing, the latest RTTOV coefficients available for visible/IR sensors are trained using LBLRTM v12.8 with the AER v3.6 molecular database

and MT-CKD_3.2 for continuum absorption. For microwave sensors the AMSUTRAN model based on MPM89 (Liebe 1989) is used, and is described fully in Turner et al. (2019). In contrast to the visible/IR case, the microwave LBL transmittance calculations are performed during the coefficient generation process with no requirement for pre-computed transmittance databases.

   RTTOV predicts optical depths due to water vapour and, optionally, a selection of other trace gases: ozone ($O_3$), carbon dioxide ($CO_2$), nitrous oxide ($N_2O$), carbon monoxide (CO), methane ($CH_4$), and sulphur dioxide ($SO_2$). To account for the

absorption by other radiatively significant gases, coefficients are also computed to obtain the optical depth due to "mixed gases" which comprise fixed climatological profiles of relevant gases that subsequently cannot be varied in the RTTOV simulations.

   In the existing parameterisation, RTTOV implements three sets of predictors, named after the RTTOV versions in which they were introduced. The "v7 predictors" (Saunders et al., 1999) predict optical depths due to mixed gases, water vapour and optionally $O_3$. They were designed primarily for satellites in low Earth orbit and as such support training for satellite

zenith angles up to about $64^{\circ}$ (the training is done for 6 evenly spaced secants from 1 to 2.25). The "v8 predictors" (Saunders et al., 2006) employed separate regressions for water vapour lines and water vapour continuum and added $CO_2$ as an optional variable gas. The "v9 predictors" (Matricardi, 2008) added $N_2O$, CO and $CH_4$ (and later, in RTTOV v12, $SO_2$) as optional variable gases, and for short-wave IR channels they were designed to support training for zenith angles up to $85^{\circ}$ (the training is done for 14 zenith angles, up to a maximum secant of 12) in order to support solar radiation. In RTTOV v11, the v9 predictors

were applied to the simulation of visible channels, and for geostationary (GEO) sensors the training was extended to the full 14 secants for all channels. The v7 and v8 predictors are the same across the whole IR spectrum, while the v9 predictors vary in different spectral bands.

   A critical aspect of this approach is the fact that the predicted optical depths are not monochromatic, but rather are "polychromatic" since they are for satellite channels of finite spectral width. These are computed by integrating the high spectral





resolution LBL transmittances over the channel spectral response functions. This means that simply summing the individual optical depths due to each gas would not yield the total optical depth due to all gases combined. To mitigate this error, the parameterisation instead predicts "effective" optical depths which are calculated from "effective" transmittances which in turn are ratios of channel-integrated transmittances following McMillin et al. (1995) as illustrated in Eq. (1) for the case of variable water vapour and $O_3$.

$$\tau_j^{total} = \tau_j^{mixed} \cdot \frac{\tau_j^{mixed+wv}}{\tau_j^{mixed}} \cdot \frac{\tau_j^{mixed+wv+o3}}{\tau_j^{mixed+wv}}, \text{ for } j = 1, n \qquad (1)$$

where $j$ is the level number, $n$ is the number of levels (typically 54 or 101 as noted above), and $\tau_j$ is the channel-integrated transmittance from space to level $j$. Effective optical depths computed from the transmittance ratios on the right-hand side of Eq. (1) are predicted by the parameterisation so that after summing the effective optical depths due to each gas, $\tau_j^{total}$ (which is identical to $\tau_j^{mixed+wv+o3}$) is obtained as required.

The existing RTTOV predictors are proven to be accurate and computationally efficient, and are widely used in a variety of operational and research applications as noted in the introduction. However, when considering future developments of RTTOV such as adding new variable gases, the effective transmittance approach for the gases has some shortcomings. It is sensitive to the order of the gases in the sequence of ratios which means it may be necessary to change this order on a per-channel basis. It is necessary to take steps to handle numerical issues caused by very small transmittances in the denominators. Adding a new variable gas can be a difficult process as it may require cross-gas predictors (e.g. predictors involving the $CO_2$ concentration may be required when calculating the effective optical depth for water vapour) because the effective transmittance for one gas is not independent of other gas concentrations. Finally, due to these complications there is a practical limit to the number of variable gases this approach can support.

It is desirable to rationalise the three sets of RTTOV predictors into a single predictor set that may be trained over a wide range of zenith angles across the full spectrum to support GEO sensors and solar-affected channels, that in principle supports any combination of variable gases, and in which it is easier to add new variable gases as required by the evolving demands of users.

## 3   The new optical depth parameterisation

The new parameterisation is based on the method described in McMillin et al. (2006). The channel-integrated layer optical depths are predicted for the mixed gases and for each variable gas independently. A final correction term is computed to account for the error due to summing polychromatic optical depths. The transmittance calculation is illustrated in Eq. (2) for the case of variable water vapour and $O_3$.

$$\tau_j^{total} = \tau_j^{mixed} \cdot \tau_j^{wv} \cdot \tau_j^{o3} \cdot \tau_j^{c}, \text{ for } j = 1, n \qquad (2)$$

where $\tau^c$ is referred to as the correction term. The predictors used for the correction term for a given channel depend upon the gases which contribute to the optical depth in that particular layer. For most gases, if any channel-integrated gas layer





optical depth among the training profile set exceeds 0.005 then predictors for that gas are included in the correction term. With variable $CH_4$ the forward model radiances and the $CH_4$ Jacobians were improved by only including the $CH_4$ predictors in the correction term if any optical depth on any level among the training profile set exceeds 0.01. The mixed gas correction term predictors are always included in the correction term. The application of these optical depth thresholds not only brings

performance benefits (excluding calculations for gases which have no impact), but more importantly significantly reduces the occurrence of spurious sensitivities to particular gases in the Jacobians. In this way the number of predictors for the correction term with non-zero coefficients varies layer-by-layer and channel-by-channel.

In the coefficient training, the correction term is calculated as shown in Eq. (3).

$$\tau_j^c = \frac{\tau_j^{total}}{\hat{\tau}_j^{mixed}\hat{\tau}_j^{wv}\hat{\tau}_j^{o3}}, \text{ for } j = 1, n \tag{3}$$

where $\hat{\tau}_j$ represents the parameterised transmittance. By using the parameterised gas transmittances in the denominator rather than the channel-integrated transmittances from the LBL model, the correction term also mitigates errors in the individual gas optical depth regressions.

The new "v13 predictors" were derived from the v9 predictors through a combination of stepwise regression and trial and error. The new predictors are given in full in Appendix A. Tables A2, A3 and A4 indicate which predictors are used for the gas

optical depth prediction, and which predictors are used for the correction term.

The training profiles and fixed pressure levels remain the same as for the existing parameterisation. The coefficients are trained using all 14 secants (zenith angles) for solar-affected channels (those with wavelengths below 5 $\mu$m) and for all channels on GEO sensors. Ordinary Least Squares linear regressions are carried out for each of the individual gas optical depths and then for the correction term. In the regression, training optical depths are omitted for layers where the transmittance due to the gas

in question from space down to the layer is less than $3 \times 10^{-6}$ (i.e. where the layer is invisible to the satellite due to absorption by the intervening atmosphere). In order to reduce the influence of layers among the training profiles that are optically deep in the atmosphere and hence have limited impact on the top of atmosphere radiance, all predictor values and training optical depths input to the regression are weighted by the square root of the product of the transmittances from space to the levels bounding the layer. Finally, where any individual predicted gas layer optical depth is less than zero, it is set to zero before the

correction term regression is computed. Similarly, where the predicted total layer optical depth (including the correction term) is less than zero, this is also set to zero.

The new parameterisation provides benefits when managing the large databases of LBL transmittances and when adding new variable gases (these are issues in the visible/IR, but do not affect the microwave). For the old parameterisation, LBL simulations are required for the total transmittances including all atmospheric constituents, and then further simulations each

time omitting an additional variable gas. In this way a transmittance database is constructed which is used to calculate the effective transmittances illustrated in Eq. (1). A separate transmittance database is required for each configuration of variable gases. When adding a new variable gas it may be necessary to run the LBL model multiple times as a number of different sets of transmittances may need to be updated within a transmittance database. Furthermore, when developing predictors for the





new variable gas, one may have to consider predictors involving other trace gases in particular spectral regions due to the fact

that the effective transmittances are not dependent on one single gas alone.

By contrast, only one transmittance database is required for the new scheme: LBL simulations are required for the transmittances due to each variable gas alone, the transmittances for the mixed gases excluding each combination of variable gases to be supported in the RTTOV simulations, and the total transmittances with each combination of variable gases varying among the training profiles. Currently for RTTOV v13, the variable gas combinations in the visible/IR are water vapour+$O_3$, water

vapour+$O_3$+$CO_2$, and all seven variable gases supported by RTTOV. The storage requirements for the LBL transmittances are therefore reduced compared to the old scheme. Adding a new variable gas in the new scheme requires the LBL model to be run three times: once to obtain transmittances for the new gas alone, a second time to obtain the mixed gas transmittances excluding this new variable gas in addition to the other variable gases, and a third time for the total transmittances with the new gas varying among the training profiles. It is then required to develop a set of predictors for the optical depths due to

the new variable gas, noting that these are completely independent of the prediction of the other variable gas optical depths. Finally, predictors related to the new gas are required for the correction term. This process is therefore somewhat simplified with the new scheme and this promises to make RTTOV more flexible for future developments in relation to the optical depth parameterisation.

## 4   Rayleigh scattering

The LBLRTM simulations used for training v9 predictor coefficients for visible and near-IR satellite channels include extinction due to Rayleigh scattering. The result is that Rayleigh extinction is included in the predicted gas optical depths and as such it is not possible to separate it from the gas absorption. However, enabling this separation as an option in RTTOV is desirable: one reason is that it enables Rayleigh multiple scattering to be included in the full multiple scattering solver in RTTOV (Hocking, 2016) which improves the accuracy of the scattering calculations, particularly in the presence of optically thick clouds (Scheck,

175  2016).

To this end, Rayleigh extinction is excluded from the LBLRTM simulations used in training v13 predictor coefficients. Instead a fast parameterisation of Rayleigh extinction is applied within RTTOV at run-time. This follows Bucholtz (1995) who provides a parameterisation of the Rayleigh volume scattering coefficient $\beta_s$ at a standard temperature $T_s$ and pressure $p_s$ as a function of wavelength, and gives the scattering coefficient $\beta$ at arbitrary temperature $T$ and pressure $p$ as:

$$\beta = \beta_s \frac{T_s}{p_s} \frac{p}{T} \qquad (4)$$

To compute the layer optical depth due to Rayleigh extinction, $\beta_s$ is averaged over the channel spectral response (this is done off-line), and the ideal gas law and hydrostatic equation are applied to Eq. (4) to obtain the layer nadir optical depth $OD$:

$$OD = \beta_s \frac{T_s}{p_s} \frac{R}{gM_{air}} \Delta p \qquad (5)$$

where $R$ is the gas constant, $g$ is acceleration due to gravity (assumed constant), $M_{air}$ is the molar mass of dry air and $\Delta p$ is

the difference in pressure across the layer. Since the Rayleigh extinction has a smooth spectral variation, we scale the nadir





optical depth by the local path length (e.g. the secant of the zenith angle) and add it to the parameterised gas layer optical depths without introducing significant "polychromatic errors".

A simple parameterisation of Rayleigh single scattering is implemented in RTTOV (Saunders et al., 2017, 2020) for fast simulations (i.e. those not modelling full Rayleigh multiple scattering).

## 5 Validation

### 5.1 Comparisons to the LBL model for the dependent profiles

The most basic validation performed for all RTTOV coefficients is to compare the RTTOV simulated brightness temperatures and reflectances to those calculated from the channel-averaged LBL transmittances for the diverse 83 profiles used to train the coefficients. In this case the LBL radiances are obtained by a single integration of the radiative transfer equation. The surface

emissivity is set to one as the aim is to evaluate the accuracy of the layer optical depth parameterisation. This comparison examines the errors due to the optical depth regression and due to the addition of polychromatic optical depths from each of the variable gases, and it is these errors that the parameterisation seeks to minimise.

Figure 1 shows plots of the bias and standard deviation for the MSG-4 SEVIRI IR channels for the v7 predictors (variable $O_3$ only) and the v9 and v13 predictors (variable $O_3$ and $CO_2$) on 54 levels. The validation is shown for the first 6 secants

used in the coefficient training (zenith angles up to about $64^\circ$). Note that the v7 predictor coefficients are trained over these 6 secants, and as such are not applicable at the larger viewing angles, while the v9 and v13 predictor coefficients are trained over the full 14 secants (zenith angles up to about $85^\circ$). The v7 and v13 predictors typically exhibit very small biases, while the v9 predictors show biases in the water vapour (6.3 and 7.3 $\mu$m) and $CO_2$ (13.4 $\mu$m) channels. It is not clear why the v9 predictors have larger biases for water vapour channels. For the $CO_2$ channel, the larger errors are mostly due to training the

coefficients on the wider range of zenith angles: the v9 predictors in the thermal IR were not optimised for this. In terms of the standard deviations, the v13 predictors compare well with the v7 predictors, especially considering they are fitting a wider range of zenith angles, and they improve on the v9 predictors in the lower peaking water vapour channel (7.3 $\mu$m) and $CO_2$ channel particularly.

In general, the v13 predictors show small biases across the spectrum when looking at the errors in the optical depth regres-

sion: this is largely due to the correction term. The separation of the absorption due to individual gases in the new predictors also tends to reduce the bias and standard deviation in spectral regions where multiple gases contribute significantly to the total absorption.

Figure 2 shows plots of the RTTOV vs LBL statistics for IASI for the v7 and v13 predictors (variable $O_3$ only) on 101 levels. The statistics are computed for the 83 training profiles over the first 6 secants used in the training. Again the v13 predictors have

very low bias across the whole spectrum and the standard deviations compare well to the older predictors for most channels. Note also that in the short-wave IR channels, the v13 predictors are trained over 14 secants to support solar radiation, while the v7 predictors are only trained over the first 6 secants. This explains the slightly larger errors compared to the old predictors for some of these channels.





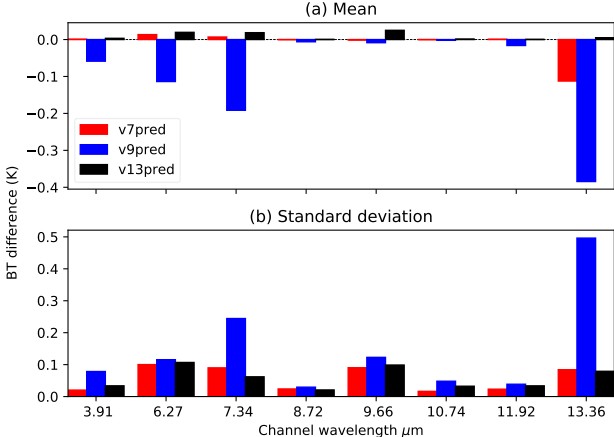

**Figure 1.** RTTOV vs brightness temperatures (BT) calculated from channel-integrated LBL transmittances for MSG-4 SEVIRI IR channels for v7 predictors (variable $O_3$), v9 predictors (variable $O_3$ and $CO_2$) and v13 predictors (variable $O_3$ and $CO_2$) for the 83 training profiles.

Figure 3 shows similar plots for IASI for the v9 and v13 predictors on 101L with all seven variable gases. Note that the optical depth prediction for high (volcanic) $SO_2$ concentrations exhibits larger errors than for other gases. This is evident in the $SO_2$ bands near 1150, 1350 and, to a lesser extent, 2500 cm$^{-1}$. The v13 predictors show significantly smaller errors compared to the v9 predictors, but this remains an aspect of the optical depth prediction to be improved in the future. It is important to note that for more typical (lower) background $SO_2$ concentrations the optical depth prediction works well (see below). Outside of the $SO_2$ bands the new predictors again show very low biases, and the standard deviations are generally comparable to or smaller than those for the old predictors.

## 5.2 Comparisons for visible and near-IR channels

Similar statistics to those presented in the previous section are produced for the visible and near-IR channels of the GOES-16 ABI sensor. In this case the simulations are carried out for a variety of satellite and solar zenith angles with a relative azimuth of $180°$ and a surface BRDF of 0.1. The top of atmosphere reflectances are computed using RTTOV, both for the parameterised optical depths and those obtained from the LBL transmittances. They therefore include the RTTOV Rayleigh single-scattering approximation, and for the v13 predictors the Rayleigh extinction parameterisation is applied to both. Figure 4 shows plots for the GOES-16 ABI visible and near-infrared channels for the v9 and v13 predictors. Here the v13 predictors mostly equal or improve on the v9 predictors, with a substantial reduction in standard deviation in the 1.37 $\mu$m channel which has strong water vapour absorption. Note that the training profile set includes some dry profiles for which this channel is surface-sensitive, and it is from these cases that we see the impact of the optical depth prediction in these statistics.

Figure 5 compares RTTOV visible/near-infrared reflectances with v9 and v13 predictors for an independent 52 profile set with a surface BRDF of 0.1 and a variety of satellite and solar zenith and azimuth angles: these simulations include the v13 predictor Rayleigh extinction parameterisation (for the v13 predictors only) and the single-scattering approximation. The largest





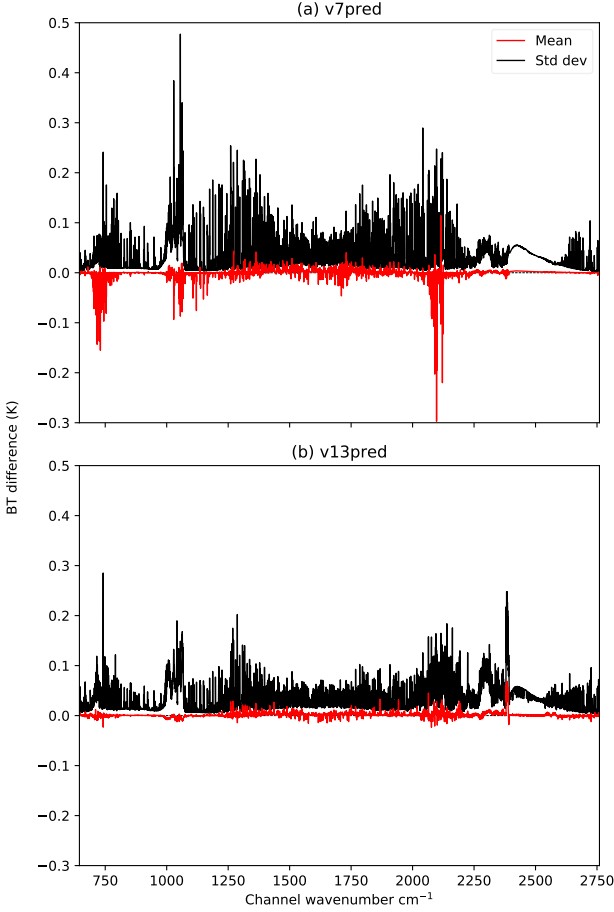

**Figure 2.** RTTOV vs brightness temperatures calculated from channel-integrated LBL transmittances for IASI (a) v7 and (b) v13 predictors with variable $O_3$ for the 83 training profiles.

differences here are again in the 1.37 $\mu$m channel, consistent with Fig. 4. For the channels below 1 $\mu$m there is a small increase

in bias with decreasing wavelength which is most likely to be due to the differences in the treatment of Rayleigh extinction. The differences are small though, and suggest that the new Rayleigh extinction parameterisation is working reasonably well. Further validation of visible and near-IR radiances is planned for the future.

### 5.3   Comparisons to the LBL model for independent profiles

It is also important to validate RTTOV against the LBL models for profile sets that are independent of the training profiles.

This section presents comparisons of RTTOV IR radiances with radiances computed from the LBLRTM model run at a spectral resolution of 0.001 cm$^{-1}$. The surface emissivity is again set to one, and the LBLRTM radiances are integrated over the channel





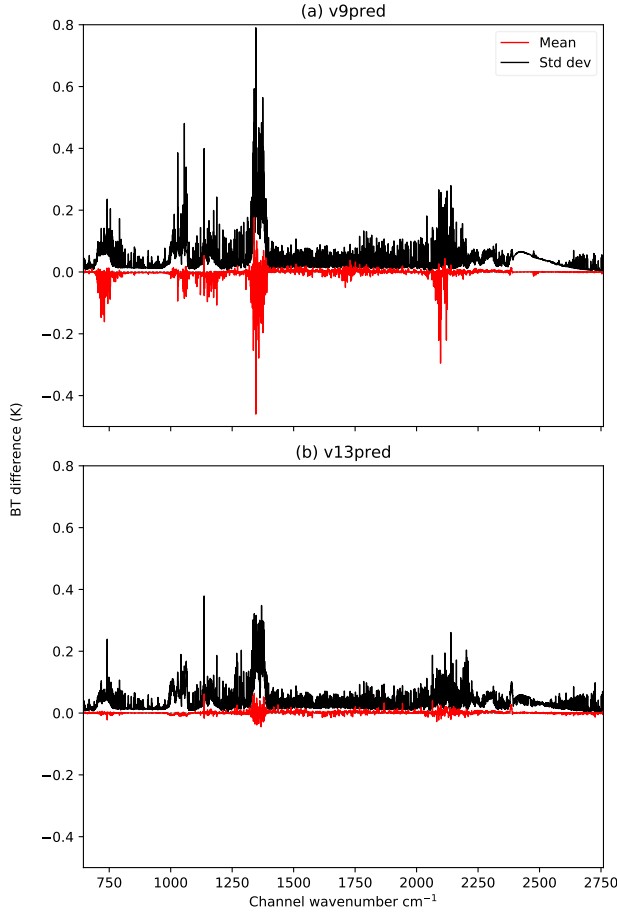

**Figure 3.** RTTOV vs brightness temperatures calculated from channel-integrated LBL transmittances for IASI (a) v9 and (b) v13 predictors with all variable gases for the 83 training profiles.

spectral response functions and compared to the RTTOV radiances. These are therefore comparisons of RTTOV to the ideal scenario of running a LBL model instead.

Figure 6 shows the differences between RTTOV and LBLRTM for the MSG-4 SEVIRI IR channels for the 5000 profiles in the temperature subset of the NWP SAF 25000 diverse profile dataset on 137 levels (Eresmaa and McNally, 2014). Simulations were run for all 14 secants used in training the coefficients for the v9 and v13 predictors with variable $O_3$ and $CO_2$. The profile dataset includes variable water vapour and $O_3$, and all other gases use fixed background profiles. While there is an increase in bias with the v13 predictors in the water vapour and $CO_2$ channels (6.3, 7.3 and 13.4 $\mu$m), the new predictors result in comparable or smaller standard deviations in all channels which is arguably more important since bias correction can mitigate larger biases.



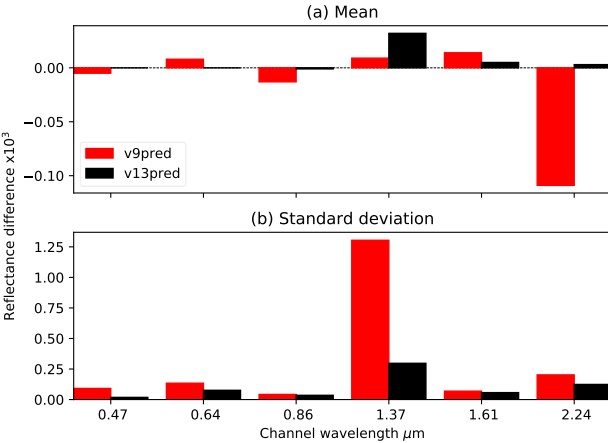

**Figure 4.** RTTOV vs reflectances calculated from channel-integrated LBL transmittances for GOES-16 ABI visible/near-IR channels for v9 and v13 predictors with variable $O_3$ and $CO_2$ for the 83 training profiles. Note that the vertical scale is magnified by a factor of 1000.

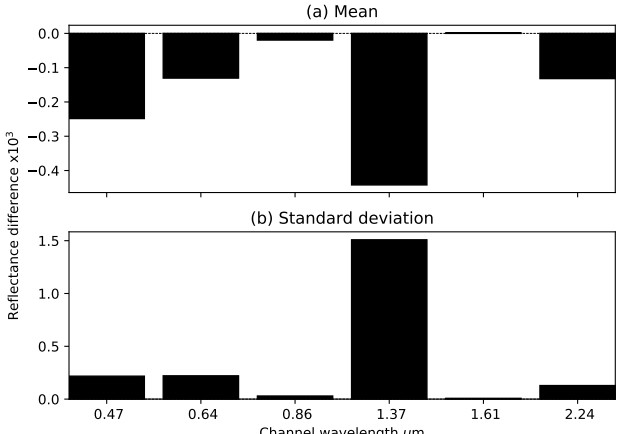

**Figure 5.** Difference in RTTOV reflectances calculated with v9 and v13 predictors with variable $O_3$ and $CO_2$ for GOES-16 ABI visible/near-IR channels for 52 independent profiles. Note that the vertical scale is magnified by a factor of 1000.

Figure 7 shows statistics for IASI for the same 5000 profiles over the first 6 secants used in coefficient training for the v7 predictors on 101 levels (variable $O_3$) and the v13 predictors on 101 levels with variable $O_3$ and $CO_2$. Standard deviations for the v13 predictors are comparable to or smaller than those for the v7 predictors across most of the spectrum, and most of the larger biases seen with the v7 predictors are eliminated with the new predictors.

Figure 8 shows the statistics for IASI for a diverse 52 profile set for the v9 predictors with six variable gases (all gases except $SO_2$) and the v13 predictors with all seven variable gases. The diverse profile set only includes variable water vapour, $O_3$ and $CO_2$ and all other gases take fixed background values. For $SO_2$ in the v13 predictors case, this is a typical low concentration



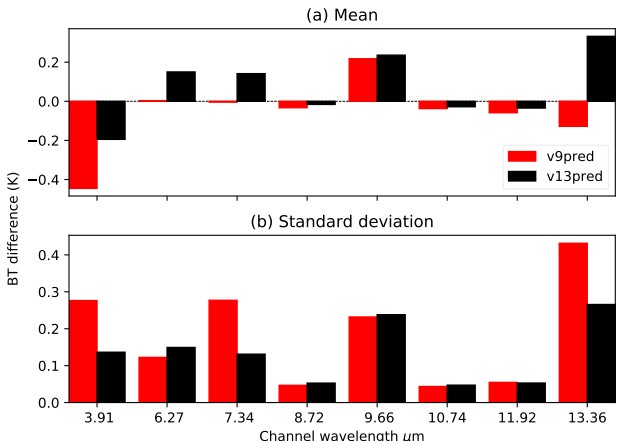

**Figure 6.** RTTOV vs brightness temperatures calculated from channel-integrated LBLRTM radiances for MSG-4 SEVIRI IR channels for v9 and v13 predictors with variable $O_3$ and $CO_2$ for 5000 independent profiles with varying water vapour and $O_3$.

profile. There is some indication of slightly increased bias and standard deviation for the v13 predictors in the 1350 cm$^{-1}$ $SO_2$ band which is related to the $SO_2$ optical depth prediction, but otherwise the new predictors with variable $SO_2$ compare
reasonably well to the v9 predictors with fixed $SO_2$. A full validation of the $SO_2$ optical depth prediction requires a suitable set of profiles from volcanic eruptions, and this planned for the future.

Finally, Figure 9 shows a comparison for the ATMS microwave sensor. This plot shows statistics comparing RTTOV with radiances computed from channel-integrated LBL transmittances for v7 and v13 predictors for all 25000 profiles in the NWP SAF diverse profile dataset on 137 levels over the 6 secants used in training. For microwave sensors, radiances computed
from channel-integrated transmittances are very similar to channel-integrated radiances from high resolution transmittances, so only the former are shown here. The differences between RTTOV and the LBL are much smaller in the microwave than in the IR because the errors due to summing polychromatic optical depths are smaller since the absorption spectrum is much more smooth in the microwave than the IR. For the temperature sounding and window channels (1-16) there is no significant difference between the old and new predictors. For the water vapour channels, there is an indication of a small degradation with
the new predictors in channels 17 (165 GHz) and 22 (the 183 GHz channel closest to the absorption line), but the differences in bias and standard deviation between the old and new predictors are well below 0.1 K.

**5.4  Jacobians**

It is important also to examine the Jacobians that RTTOV computes from the predictor scheme. Saunders et al. (2018) showed how the v7 predictor temperature and water vapour Jacobians agree well with Jacobians computed from the AMSUTRAN
LBL model. Figure 10 shows the mean temperature and water vapour Jacobians from the v7 and v13 predictors for ATMS calculated over the 83 profiles used for training RTTOV. In general the differences in both the temperature and water vapour

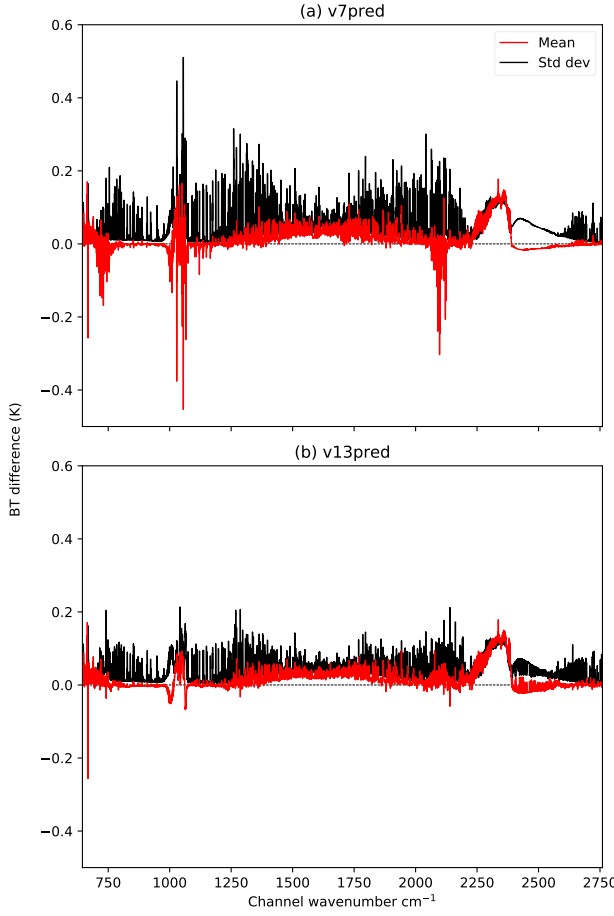

**Figure 7.** RTTOV vs brightness temperatures calculated from channel-integrated LBLRTM radiances for IASI (a) v7 predictors (variable v13 $O_3$) and (b) v13 predictors (variable $O_3$ and $CO_2$) for 5000 independent profiles with varying water vapour and $O_3$.

Jacobians between the old and new predictors are very small. A subset of channels are plotted for clarity: for the other channels the differences are smaller than those shown.

Figure 11 shows the mean temperature and water vapour Jacobians for a selection of IASI channels representing different parts of the spectrum. In this case the v9 and v13 predictor coefficients with all trace gases are used. The Jacobians show slightly larger differences than for ATMS, but overall the shape and magnitudes of the Jacobians are similar between the old and new predictors.

Further validation of the Jacobians is planned including testing in full operational data assimilation systems.

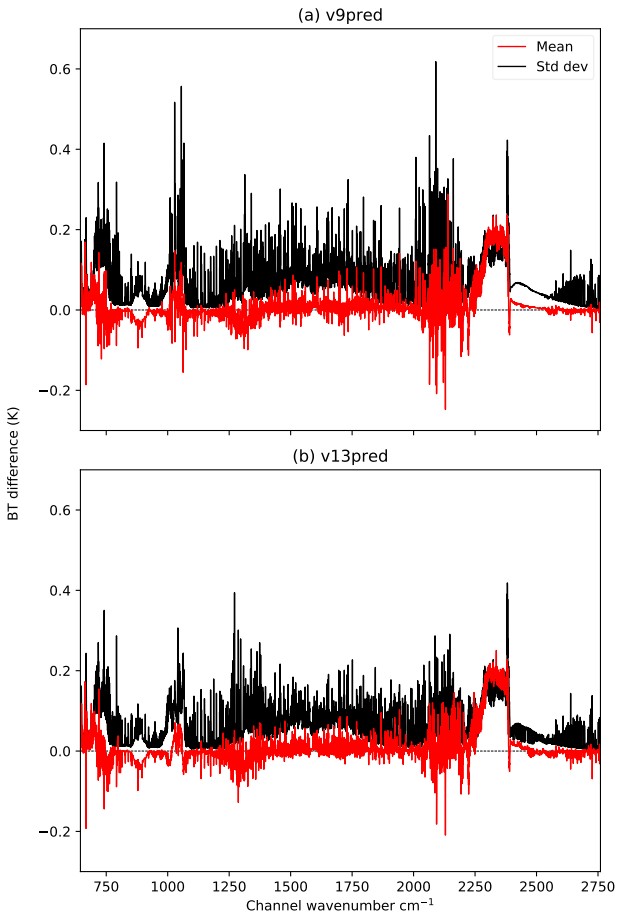

**Figure 8.** RTTOV vs brightness temperatures calculated from channel-integrated LBLRTM radiances for IASI (a) v9 predictors (all gases except $SO_2$) and (b) v13 predictors (all seven variable gases) for 52 independent profiles with varying water vapour, $O_3$ and $CO_2$.

## 5.5 Validation in an operational NWP assimilation system

Infrared and microwave measurements of spectral radiances made between 1 March 2020 and 31 March 2020 are compared with simulations performed using the RTTOV v13 model with the latest regression coefficients files available.

Monitoring experiments, which examine changes in first guess departures without generating a new analysis and forecast, were conducted within the framework of the ECMWF (European Centre for Medium-Range Weather Forecasts) Integrated Forecasting System using model fields of temperature, water vapour and ozone obtained from short-range forecasts. This

ensures that a wide range of atmospheric scenarios are sampled and enables the examination of the change in the simulated brightness temperature due to a change in the observation operator only and not through subsequent changes in the analysis field that would result from a full-cycling data assimilation system. All experiments are based on cycle 47R1 of the operational system and use the same parent experiment, but have been run at the lower spatial model resolution of $T_{CO}$ 399 (approximately



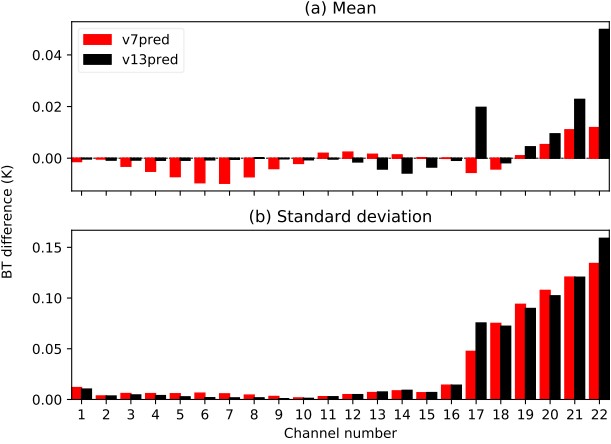

**Figure 9.** RTTOV vs brightness temperatures calculated from channel-integrated LBL transmittances for ATMS v7 and v13 predictors for 25000 independent profiles.

25 km) with 137 levels in the vertical. The statistics presented here are computed before bias correction is applied to the observations.

The impact of using RTTOV v13 with the new v13 predictors for microwave sounders was studied and comparisons against v7 predictors performed. Figure 12 shows observation minus first guess (before bias correction) statistics comparisons between v7 and v13 predictors for ATMS onboard S-NPP and NOAA-20. As expected, using the v13 predictors has a rather small impact on the ATMS brightness temperatures simulated by RTTOV v13 in both temperature sounding channels 6–15 and humidity sounding channels 18–22. Mean biases using v13 predictors lies within the range of biases observed with v7 predictors (Fig. 12a). For the 183 GHz humidity channels, using v13 predictors leads to a small reduction of the standard deviation of first guess departures, as seen in channel 22 where the difference in standard deviation between the v13 and v7 predictors can reach up to 0.01 K (Fig. 12c). For the temperature channels, the effect of the v13 predictors on the standard deviations is smaller than for the humidity sounding channels, but consistent with changes seen for AMSU-A instruments used in the ECMWF system (not shown).

Experiments have also been performed for the new RTTOV coefficient files trained on LBLRTM v12.8 for infrared sensors. Figure 13 shows the comparisons between the RTTOV v13 computed first guess departures for water vapour channels on geostationary radiances using the v7 and v13 predictor coefficients based on LBLRTM v12.8 and with variable $O_3$ only. The new v13 predictors exhibit slightly larger biases for all geostationary satellite water vapour channels (Fig. 13a). The v13 predictors results compare favourably with the v7 predictors results in terms of standard deviation. The difference in standard deviation of first guess departures between v13 and v7 $O_3$ only predictors are below 0.01 K for water vapour channels on geostationary satellites (Figs. 13b-c).

In the following we present an evaluation of hyperspectral infrared radiance from IASI onboard MetOp-A/B/C in terms of departure statistics against clear-sky brightness temperatures simulated from short-term forecasts. Three monitoring experi-



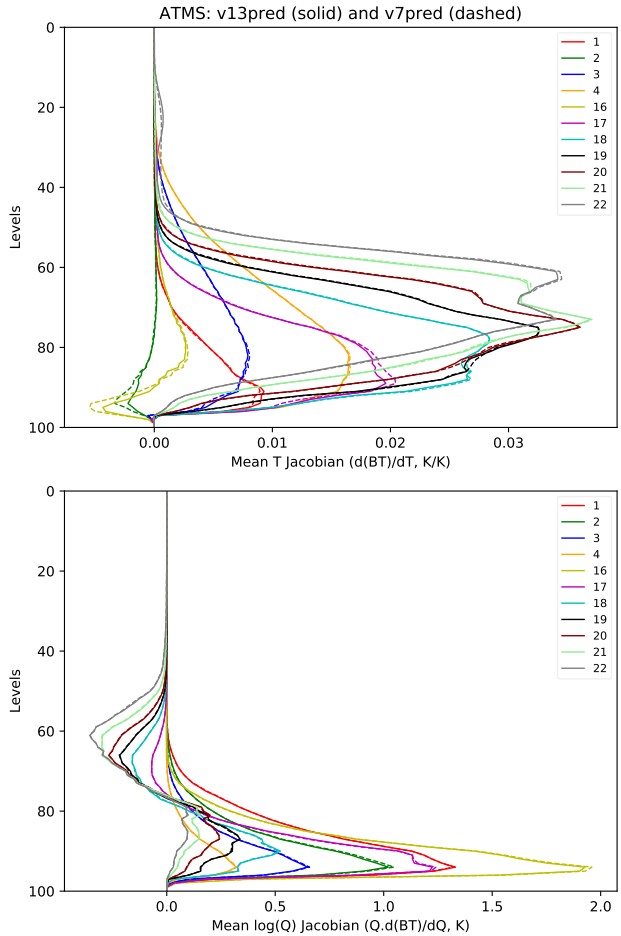

**Figure 10.** Mean ATMS temperature (top) and water vapour (bottom) Jacobians for v13 predictors (solid lines) and v7 predictors (dashed lines) for the 83 training profiles. A subset of channels are shown for which the differences are largest.

ments have been carried out to compare IASI observed radiances to radiances simulated by RTTOV v13: once using v13 and v7 predictors with variable $O_3$ on 101 levels (Fig. 14), once using v13 and v8 predictors with variable $O_3$ + $CO_2$ on 101 levels (Fig. 15) and finally using v13 predictors with all seven variable gases and the v9 predictors with six variable gases (excluding $SO_2$) on 101 levels (Fig. 16).

Figures 14–16 evaluate IASI channel performance in terms of mean (a-b) and standard deviation (c-d) of first guess departure

before bias correction, shown as a function of channel central wavenumber in band 1 (645–1200 cm$^{-1}$) and band 2 (1200–2000 cm$^{-1}$). The statistics have been evaluated over a one-month period (March 2020) to ensure the adequate representation of channels that are frequently removed because of cloud contamination. For all experiments, results show very close similarities between the observation minus first guess bias calculated with infrared coefficients files based on the new v13 predictors or the v7/8/9 predictors. In band 1 and 2, the mean first guess departure rarely exceeds 0.5 K. The noted exception is the ozone



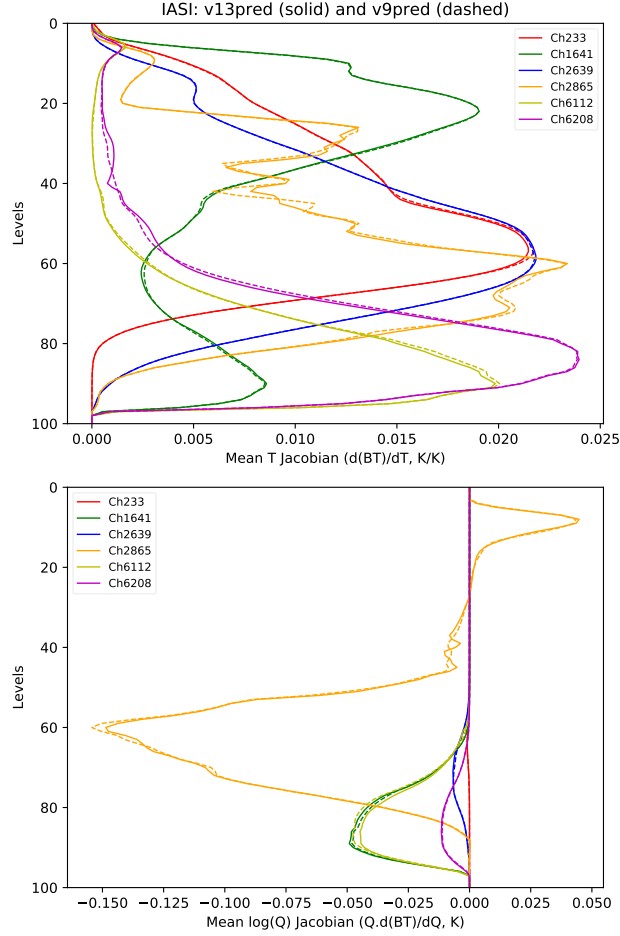

**Figure 11.** Mean IASI temperature (top) and water vapour (bottom) Jacobians for v13 predictors (solid lines) and v9 predictors (dashed lines) for the 83 training profiles.

band (1000–1080 cm$^{-1}$) where biases can reach 1 K when RTTOV v13 coefficient files based on v7/8/9 predictors are used. The standard deviation of first guess departures is consistently less than 0.4 K outside the water-vapour absorption lines where it varies in the range 1–1.3 K.

In Fig. 17 we have plotted the difference in the standard deviation of observation minus first guess departures between v13 and v7/8/9 predictors, respectively. A reduction in the standard deviation of the differences can be used as a measure of the improvement of the radiative transfer model performance if only the radiative transfer model has changed. Notable differences are seen in correspondence with the O$_3$ absorption lines at 1000–1080 cm$^{-1}$, where the new v13 predictors appear to better separate the contribution of each molecule to the total transmittance, resulting in a reduction in standard deviation of up to 0.2 K in some channels when compared with v7/8/9 results. Smaller differences (less than 0.03 K) are seen in terms of the standard deviations at 670–770 cm$^{-1}$. The version v13 predictors appear more accurate than v9 predictors for sounding



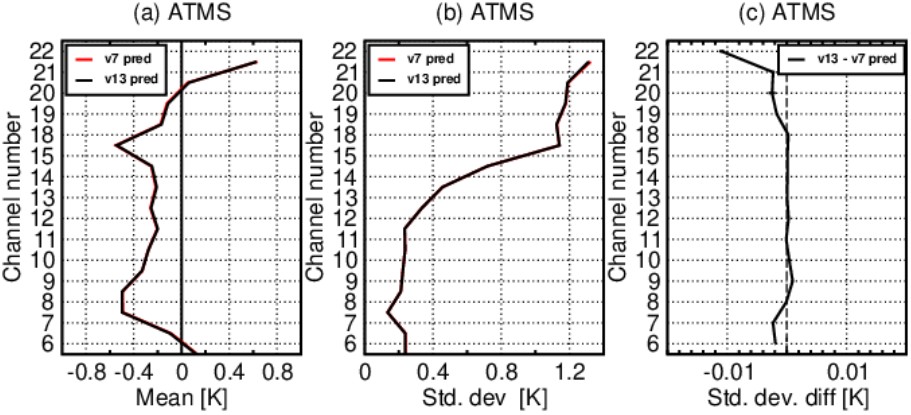

**Figure 12.** Global statistics of observation minus background (O-B) departure before variational bias correction for ATMS on S-NPP/NOAA-20 during March 1–31, 2020 in the ECMWF system: (a) Mean (O-B); (b) Standard deviation of (O-B); (c) Difference of the standard deviations; ATMS statistics for the v13 predictors experiment are shown in black, whereas statistics for the v7 predictors experiment are shown in red.

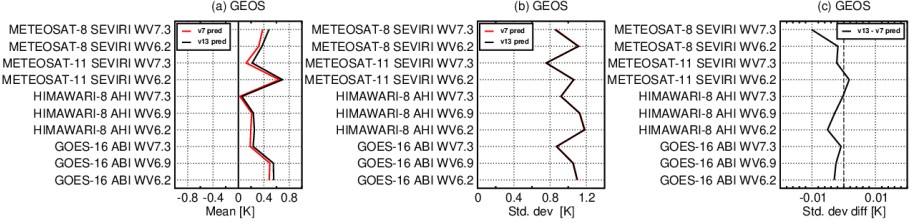

**Figure 13.** Global statistics of observation minus background (O-B) departure before variational bias correction for geostationary radiances during March 1–31, 2020 in the ECMWF system: (a) Mean (O-B); (b) Standard deviation of (O-B); (c) Difference of the standard deviations; Statistics for the v13 predictors with variable $O_3$ experiment are shown in black, whereas statistics for the v7 predictors with variable $O_3$ experiment are shown in red.

channels sensitive to tropospheric temperature in the wavenumber range 710–770 $cm^{-1}$, but the v7 predictors are better than v13 predictors for the same region of the spectrum. However the differences in standard deviation between the old and new predictors in this region are well below the IASI instrument noise in these channels (for example, see Fig. 1 in Crevoisier et al. (2014)).

     Results from these initial experiments are encouraging when examined in terms of departure statistics against clear-sky

brightness temperatures simulated from short-term forecasts as used in ECMWF's 4D-Var assimilation system for a variety of infrared and microwave instruments. Further examinations are needed to assess the accuracy of the coefficients based on the new v13 predictors and evaluate their benefit in an assimilation context.





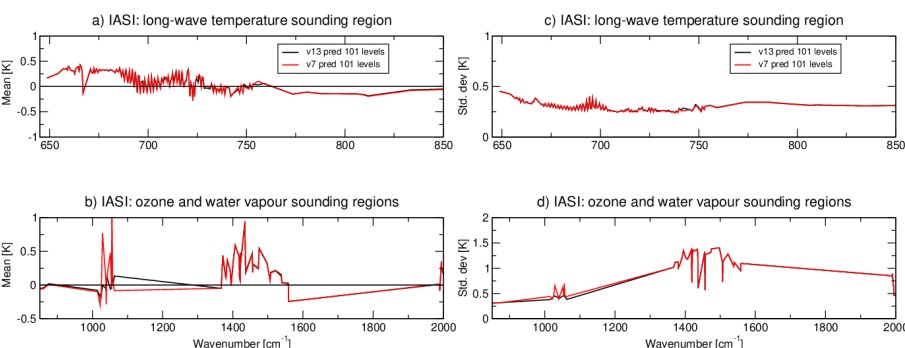

**Figure 14.** The mean and standard deviation values of the difference between observed and simulated brightness temperatures over the globe for a selection of IASI channels on MetOp-A/B/C in band 1 (a) and band 2 (b). Calculations are performed with RTTOV v13 with coefficients that are based on the new v13 predictors and the v7 predictors (variable $O_3$ only on 101 levels).

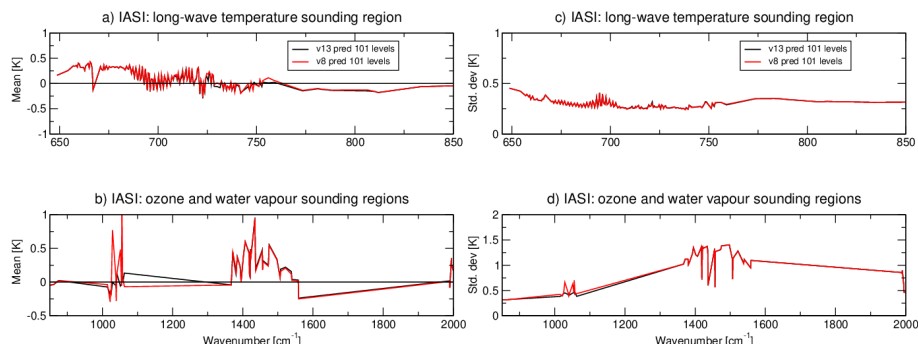

**Figure 15.** As Fig. 14, but with RTTOV v13 coefficients that are based on the new v13 predictors and the v8 predictors (variable $O_3$ and $CO_2$ on 101 levels).

## 6 Conclusions

This paper presents a new optical depth parameterisation which has been implemented for the recent major release of RTTOV,
version 13. The new parameterisation provides a single set of predictors that can be used for any combination of variable gases across the spectrum from the visible to the microwave. The coefficients can be trained over a wide range of zenith angles across the full spectrum to support geostationary sensors and simulations including solar radiation. The new parameterisation can be extended to additional variable gases more easily than the existing scheme.

Validation by comparison to line-by-line simulations indicates that the new parameterisation equals or reduces errors in
the optical depth regression compared to the existing RTTOV parameterisation in most spectral regions. Comparisons to full line-by-line radiances for independent profiles indicates the accuracy is comparable to the existing parameterisation for microwave and broadband infrared radiometers, and may improve the accuracy for hyperspectral IR sounders. The Jacobians are



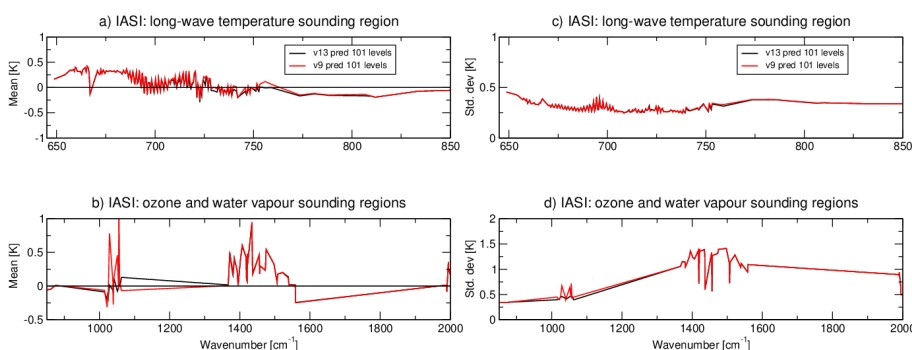

**Figure 16.** As Fig. 14, but with RTTOV v13 coefficients that are based on the new v13 predictors with all seven variable gases and the v9 predictors with six variable gases (all gases except $SO_2$).

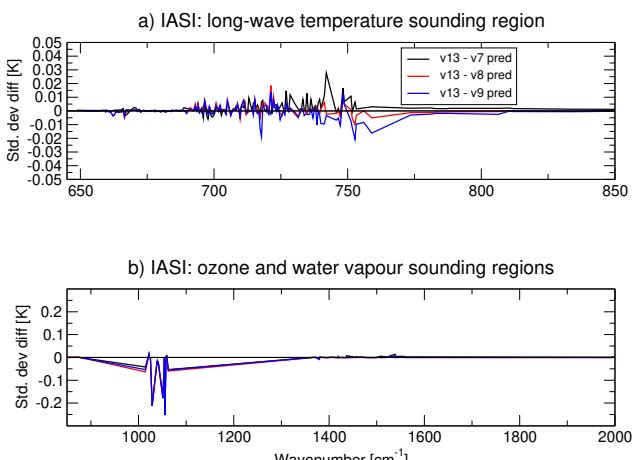

**Figure 17.** For selected IASI channels in band 1 (a) and band 2 (b), the difference of the standard deviations of the brightness temperature fit between: v13 predictors and the v7 predictors with variable $O_3$ (black line), v13 predictors and the v8 predictors with variable $O_3$ and $CO_2$ (red line) and v13 predictors with all seven variable gases and the v9 predictors with six variable gases (blue line).

very similar to those obtained from the existing parameterisation and as such no problems are expected when using the new predictors in retrieval and assimilation applications.

Results from monitoring experiments in the ECMWF operational NWP system are generally positive, showing similar or reduced standard deviations with the v13 predictors compared to the old v7/8/9 predictors for a variety of microwave and infrared sensors. In particular there is a notable reduction in standard deviation of up to 0.2 K with the new predictors for IASI in the ozone band around 1050 cm$^{-1}$. This result will be further evaluated at other NWP centres.

The new predictors are expected to be more computationally expensive than the old parameterisation due to the additional

cost of the correction term calculation. Testing indicates that the new predictors are up to 30% slower for the direct and tangent linear models, and up to 20% slower for the adjoint and Jacobian models, although there is substantially variability depending





on the compiler and the type of simulation (for example, the number of variable gases). In the context of an operational data assimilation system this increase should not present significant problems.

Future work will involve validation of the accuracy of the new predictors in operational applications such as NWP assim-
ilation systems, as well as off-line studies comparing RTTOV with other radiative transfer models such as CRTM. This will include further validation of visible and near-infrared radiances including Rayleigh scattering, the optical depth parameterisation for all variable trace gases, and the Jacobians for temperature and all variable gases. It is planned to extend the RTTOV spectral range to cover the ultra-violet in support of sensors such as Sentinel 4 and 5. This will involve investigating the application of the new predictors to this new spectral region and the possible inclusion of new variable gas species such as
nitrogen dioxide ($NO_2$). The parameterisation will also be extended by the addition of new variable gases in support of user requirements as they arise.

*Code and data availability.*    The RTTOV model can be downloaded free of charge from the NWP SAF website (https://nwp-saf.eumetsat.int, last access: 16 November 2020) once users have registered on the site to agree to the licence conditions. Updates to the code and coefficients for new instruments are also posted on the site. RTTOV v13.0 was released in November 2020 and coefficients based on the new v13
predictors as well as the older predictor versions are available for download from the website. The website also hosts plots of statistics of LBL vs RTTOV comparisons over the training profiles similar to those discussed in this paper for the majority of RTTOV coefficient files for all predictor versions (https://nwp-saf.eumetsat.int/site/software/rttov/download/coefficients/comparison-with-lbl-simulations/).

## Appendix A:  v13 predictors

The tables in this section detail the predictors used for the gas optical depth and the correction term regressions.



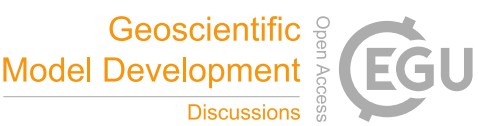

**Table A1.** Quantities used in predictor calculations listed in Tables A2, A3 and A4

| | | |
|---|---|---|
| $p_{\delta p}(j) = p(j+1)(p(j+1) - p(j))$ | $p_{\delta p}(0) = p(1)(p(2) - p(1))$ | |
| $T(j) = \frac{1}{2}(T^{\mathrm{prof}}(j) + T^{\mathrm{prof}}(j+1))$ | $T^*(j) = \frac{1}{2}(T^{\mathrm{ref}}(j) + T^{\mathrm{ref}}(j+1))$ | $\delta T(j) = T(j) - T^*(j)$ |
| $G(j) = \frac{1}{2}(G^{\mathrm{prof}}(j) + G^{\mathrm{prof}}(j+1))$ | $G^*(j) = \frac{1}{2}(G^{\mathrm{ref}}(j) + G^{\mathrm{ref}}(j+1))$ | |
| $T_r(j) = T(j)/T^*(j)$ | $T_w(j) = \dfrac{\sum_{i=1}^{j} p_{\delta p}(j-1)T(j)}{\sum_{i=1}^{j} p_{\delta p}(j-1)T^*(j)}$ | |
| $G_r(j) = G(j)/G^*(j)$ | $G_w(j) = \dfrac{\sum_{i=1}^{j} p_{\delta p}(j-1)G(j)}{\sum_{i=1}^{j} p_{\delta p}(j-1)G^*(j)}$ | $G_{wt}(j) = \dfrac{\sum_{i=1}^{j} p_{\delta p}(j-1)T(j)G(j)}{\sum_{i=1}^{j} p_{\delta p}(j-1)T^*(j)G^*(j)}$ |

All quantities defined in the table above are on layers $j$ bounded by levels $j$ and $j+1$ (aside from $p_{\delta p}(0)$), where level 1 is at the top of the atmosphere.

$p(j)$ is the pressure (hPa) at level $j$. These are usually the pre-specified pressures of the 54 or 101 levels used for RTTOV coefficients.

$T^{\mathrm{prof}}(j)$ is the temperature (K) at level $j$ of the input profile.

$T^{\mathrm{ref}}(j)$ is the temperature (K) at level $j$ of the reference profile which is the mean over the training profile set.

$G \in \{W, O3, CO2, N2O, CO, CH4, SO2\}$ represents gas concentration (ppmv over dry air).

$G^{\mathrm{prof}}(j)$ are the gas concentrations at level $j$ of the input profile.

$G^{\mathrm{ref}}(j)$ are the gas concentrations at level $j$ of the reference profile which is the mean over the training profile set.

In Tables A2, A3 and A4, $\theta$ is the zenith angle.



**Table A2.** Mixed gas and water vapour predictors and predictor lists for optical depth and correction term

| Predictor | Mixed gases | Water vapour lines | Water vapour continuum |
|:---:|:---:|:---:|:---:|
| 1 | $\sec(\theta)$ | $(\sec(\theta)W_r)^2$ | $\sec(\theta)W_r^2/T_r$ |
| 2 | $\sec^2(\theta)$ | $\sec(\theta)W_w$ | $\sec(\theta)W_r/T_r$ |
| 3 | $\sec(\theta)T_r$ | $(\sec(\theta)W_w)^2$ | $\sec(\theta)W_r^2/T_r^4$ |
| 4 | $\sec(\theta)T_r^2$ | $\sec(\theta)W_r\delta T$ | $\sec(\theta)W_r/T_r^2$ |
| 5 | $T_r$ | $\sqrt{\sec(\theta)W_r}$ | - |
| 6 | $T_r^2$ | $\sqrt[4]{\sec(\theta)W_r}$ | - |
| 7 | $\sec(\theta)T_w$ | $\sec(\theta)W_r$ | - |
| 8 | $\sec(\theta)T_r^3$ | $(\sec(\theta)W_w)^{1.5}$ | - |
| 9 | $\sec(\theta)\sqrt{\sec(\theta)T_r}$ | $(\sec(\theta)W_r)^{1.5}$ | - |
| 10 | $1$ | $(\sec(\theta)W_r)^{1.5}\delta T$ | - |
| 11 | - | $\sqrt{\sec(\theta)W_r}\delta T$ | - |
| 12 | - | $(\sec(\theta)W_w)^{1.25}$ | - |
| 13 | - | $\sec(\theta)W_r^2/W_w$ | - |
| 14 | - | $\sqrt{\sec(\theta)W_r}W_r/W_{wt}$ | - |
| 15 | - | $\sec(\theta)\sqrt{W_w}$ | - |
| Optical depth | 1-9 | 1-13 / 1-14* | 1-4 |
| Correction term | 2, 3, 4, 10 | 2, 4, 5, 6, 15 | -† |

* 1-13 for $\nu <= 1095 cm^{-1}$ or $2320 < \nu <= 2570 cm^{-1}$, otherwise 1-14, where $\nu$ is the channel central wavenumber

† No correction term predictors for water vapour continuum.





**Table A3.** $O_3$, $CO_2$, $N_2O$ predictors and predictor lists for optical depth and correction term

| Predictor | $O_3$ | $CO_2$ | $N_2O$ |
|-----------|-------|--------|--------|
| 1 | $\sec(\theta)O3_r$ | $\sec(\theta)CO2_r$ | $\sec(\theta)N2O_r$ |
| 2 | $\sqrt{\sec(\theta)O3_r}$ | $T_r^2$ | $\sqrt{\sec(\theta)N2O_r}$ |
| 3 | $\sec(\theta)O3_r\delta T$ | $\sec(\theta)T_r$ | $\sec(\theta)N2O_r\delta T$ |
| 4 | $\sec(\theta)O3_r/O3_w$ | $\sec(\theta)T_r^2$ | $(\sec(\theta)N2O_r)^2$ |
| 5 | $(\sec(\theta)O3_r)^2$ | $T_r$ | $N2O_r\delta T$ |
| 6 | $\sec(\theta)O3_r^2 O3_w$ | $\sec(\theta)T_w$ | $\sqrt[4]{\sec(\theta)N2O_r}$ |
| 7 | $\sqrt{\sec(\theta)O3_r}O3_r/O3_w$ | $(\sec(\theta)CO2_w)^2$ | $\sec(\theta)N2O_w$ |
| 8 | $\sec(\theta)O3_r O3_w$ | $\sec(\theta)T_w\sqrt{T_r}$ | $\sec(\theta)N2O_{wt}$ |
| 9 | $(\sec(\theta)O3_w)^{1.75}$ | $\sqrt{\sec(\theta)CO2_r}$ | $\sqrt{\sec(\theta)N2O_r}N2O_r/N2O_w$ |
| 10 | $\sec(\theta)O3_r\sqrt{\sec(\theta)O3_w}$ | $T_r^3$ | $(\sec(\theta)N2O_{wt})^2$ |
| 11 | $(\sec(\theta)O3_w)^2$ | $\sec(\theta)T_r^3$ | $(\sec(\theta)N2O_{wt})^3$ |
| 12 | $\sqrt{\sec(\theta)}O3_w^2\delta T$ | $\sqrt{\sec(\theta)}T_r^2 T_w^3$ | $\sec^2(\theta)N2O_{wt}\delta T$ |
| 13 | $\sec(\theta)O3_w$ | $T_r^2 T_w^2$ | - |
| 14 | - | $\sec(\theta)CO2_w$ | - |
| Optical depth | 1-12 | 1-13 | 1-12 |
| Correction term | 13 | 14, 8, 9 | 7, 8, 10, 11, 12 |





**Table A4.** CO, CH$_4$, SO$_2$ predictors and predictor lists for optical depth and correction term

| Predictor | CO | CH$_4$ | SO$_2$ |
|---|---|---|---|
| 1 | $\sec(\theta)CO_r$ | $\sec(\theta)CH4_r$ | $(\sec(\theta)SO2_r)^2$ |
| 2 | $\sqrt{\sec(\theta)CO_r}$ | $\sqrt{\sec(\theta)CH4_r}$ | $\sec(\theta)SO2_w$ |
| 3 | $\sec(\theta)CO_r\delta T$ | $\sec(\theta)CH4_r\delta T$ | $(\sec(\theta)SO2_w)^2$ |
| 4 | $(\sec(\theta)CO_r)^2$ | $(\sec(\theta)CH4_r)^2$ | $\sec(\theta)SO2_r\delta T$ |
| 5 | $\sqrt{\sec(\theta)CO_r}\delta T$ | $CH4_r\delta T$ | $\sqrt{\sec(\theta)SO2_r}$ |
| 6 | $\sqrt[4]{\sec(\theta)CO_r}$ | $\sqrt[4]{\sec(\theta)CH4_r}$ | $\sqrt[4]{\sec(\theta)SO2_r}$ |
| 7 | $\sec(\theta)CO_r\delta T\lvert\delta T\rvert$ | $\sec(\theta)CH4_{wt}$ | $\sec(\theta)SO2_r$ |
| 8 | $\sec(\theta)CO_r^2/CO_w$ | $CH4_{wt}$ | $\sqrt{\sec(\theta)SO2_r}SO2_r/SO2_{wt}$ |
| 9 | $\sqrt{\sec(\theta)CO_r}CO_r/CO_w$ | $(\sec(\theta)CH4_w)^2$ | $(\sec(\theta)SO2_w)^{1.5}$ |
| 10 | $\sec(\theta)CO_r^2/\sqrt{CO_w}$ | $\sec(\theta)CH4_w$ | $(\sec(\theta)SO2_r)^{1.5}$ |
| 11 | $(\sec(\theta)CO_w)^{0.4}$ | $\sqrt{\sec(\theta)CH4_r}CH4_r/CH4_w$ | $(\sec(\theta)SO2_r)^{1.5}\delta T$ |
| 12 | $\sqrt[4]{\sec(\theta)CO_{wt}}$ | $(\sec(\theta)CH4_w)^{1.25}$ | $\sqrt{\sec(\theta)SO2_r}\delta T$ |
| 13 | $\sec^2(\theta)CO_rCO_w$ | - | $(\sec(\theta)SO2_w)^{1.25}$ |
| 14 | $\sec(\theta)CO_w$ | - | $\sec(\theta)SO2_r^2/SO2_w$ |
| 15 | $\sec(\theta)CO_{wt}$ | - | $\sec(\theta)\sqrt{SO2_w}$ |
| 16 | $(\sec(\theta)CO_w)^2$ | - | - |
| Optical depth | 1-13 | 1-11 | 1-14 |
| Correction term | 12, 14, 15, 16 | 7, 9, 10, 12 | 2, 4, 5, 6, 15 |

*Author contributions.* James Hocking developed the new predictors and wrote much of the manuscript. Jérôme Vidot, Pascal Brunel and Pascale Roquet ran the LBLRTM simulations and generated the new visible/infrared v13 predictor coefficients. Bruna Silveira ran the validation simulations on independent profiles for the infrared sensors. Emma Turner ran the microwave line-by-line simulations. Cristina Lupu ran the monitoring experiments evaluating the new predictors in the ECMWF operational system and wrote the corresponding section of the manuscript. All authors reviewed the manuscript.





*Competing interests.* The authors declare that they have no conflict of interest.

*Acknowledgements.* RTTOV development is funded by EUMETSAT within the context of the NWP SAF.



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
