# Peer review of "A new gas absorption optical depth parameterisation for RTTOV v13"

_Geoscientific Model Development, 2020_

## Author Response (AR1)

**Reviewer 1 comments (italics) and proposed responses (bold)**

*The manuscript presents a thorough investigation and explanation of a new optical depth parameterization for a fast radiative transfer model. Â  The presentation is very well organized, and particularly appreciate the explanations in sections 2 and 3 which helped to frame the later evaluations. Â  The section on the Rayleigh scattering was a bit brief, but as eluded to later in the manuscript the in-depth exploration will continue and be presented in the future. Â  There were a lot of figures, and many with subtle differences; however, the chosen journal can support these easily and they do help to convey the evolution and small differences which are being scrutinized. Â  In short I find very little of substance which needs to be changed and would recommend publication in its current form. Â  I do have some minor comments, for a couple situations where rewording could be considered.*

**Many thanks for your comments!**

*Page 7, line 217 last sentence of the paragraph considering figure 2.   "… for some of these channels†could be changed to "such as for those with wavenumber greater than 2200 cm^-1â€.*

**The text "for some of these channels" has been changed to "for channels with wavenumber greater than 2200 cm^-1."**

*Page 10, line 254 final sentence of paragraph considering figure 6.   May want to consider a rewording here as well, maybe something like "arguably more important as data assimilation system often apply a bias correction to radiance which can mitigate these biases.â€*

**The text in the final sentence has been changed to "...arguably more important as data assimilation systems often apply bias corrections to radiances which can mitigate these larger biases."**

*Page 18 in reference to figures 12c and 13c, really just a comment. Â  Â  But these NWP result is a nice one, particularly considering the slight rise in std. dev. seen in figures 9b and 1b.*

**Agreed, the monitoring results are encouraging for the new predictors.**

*Page 20, line 365 with respect to the increase in the computation expense, this is the first mention of this in the manuscript that I recall. Â  A minor point, and I have no issue with any decision made, but would consider stating some of this earlier as RTTOV being a fast model the computational burden is always critical. Â  I agree the advancements shown with the v13 are significant and the increase is minor for a full cycling NWP and data assimilation system. Â  In particular the ability to augment and increase consideration of additional gases will likely continue to be important with current and future hyperspectral sensors. Â  Also the ability to more rapidly integrate more species with the new parametrisation seems to be of particularly large benefit.*

**We have added the following paragraph at the end of section 3 (describing the new parameterisation):**

**"The benefits of the new predictors come at a modest increase in computational cost. For example, the additional calculations required by the correction term can increase forward model run-times for clear-sky simulations by up to about 30% compared to the existing predictors. However, such an increase is not expected to be problematic for operational data assimilation systems. Furthermore, for more computationally expensive simulations such as those including cloud scattering, the relative increase in run-time is lower as the gas optical depth prediction takes a smaller proportion of the overall run-time."**
* * *
**Reviewer 2 comments (italics) and proposed responses (bold)**

*Summary: A well-written paper that elucidates the key issues associated with transmittance regression training in the RTTOV model. Some elements are missing that harm the reproducability aspect, particularly regarding the "correction term" and how exactly that is implemented. I have some minor comments below, and some questtions that could be answered in the text in a future revision.*

**Many thanks for your comments! If you have some specific suggestions regarding the description of the correction term, we would be happy to update the text. The implementation closely follows the description in McMillin et al (2006), the primary difference being that the RTTOV optical depth prediction scheme operates on fixed pressure levels instead of on a grid based on fixed absorber amounts.**

*Minor issues / formatting:*

*Equation 1: formatting needs improvement. If LaTeX formulation is being used in the equation, enclosing the /mathrm tag will make the text non-italic.*

*e.g., /mathrm{mixed}, consider /mathrm{O}_3 as well*

**All the equations have been updated to avoid italics for the text labels.**

*Throughout: "i.e." should have a comma after it, unless there's a specific style requirement here that I'm not aware of. First seen on line 140.*

**This might be a difference in convention between US and British English. In any case, commas have been added after "i.e.", and also for instances of "e.g." to be consistent.**

*Figures 13 and 16: could be larger for viewability*

**Fig 13 has been modified so that the satellite/channel labels appear only once, and the overall size is increased. Figures 13-16 have been enlarged.**

*Table A3 can be cleaned up with regard to italicization of species, subscripts etc.*

**We're not very keen on using subscripts for O3, CO2, etc in the predictor definitions in Tables A2-4 because there are already many other subscripts and in this context the "O3", for example, is a variable name representing concentration of ozone rather than being the chemical symbol for a molecule of ozone. However, we have removed the italicisation of the species as this is more consistent with usage elsewhere.**

*Technical elements:*

*Line 144 - 146:*

*"Finally, where any individual predicted gas layer optical depth is less than zero, it is set to zero before the correction term regression is computed. Similarly, where the predicted total layer optical depth (including the correction term) is less than zero, this is also set to zero."*

*Does this "truncation" introduce biases into the regression correction term?*

**The negative optical depths can occur when layer optical depths are close to zero (for individual gases or for all gases together) and the regression can then sometimes yield small negative values. We can see now that the text may be misleading. In case there is a misunderstanding it is important to clarify that the predicted correction term itself is not modified and as such no bias is introduced. We ensure that the total layer optical depth is non-negative: the total layer optical depth is the sum of the predicted correction term (which may be positive or negative) and the predicted gas layer optical depths. Before computing the correction term, we ensure the individual predicted gas layer optical depths are non-negative: this is not strictly mandatory as the correction term could theoretically account for such errors, but it seems better to deal in physically realistic optical depths rather than unphysical "artefacts" coming from the regression. The text has been clarified as follows:**

**"Similarly, where the predicted total layer optical depth is less than zero, this is also set to zero. The correction term itself may be positive or negative and as such the value computed from the regression is unmodified."**

*Regarding excluding the Rayleigh scattering calculation from LBLRTM, I wonder how this choice impacts UV channel simulations for future expansions? Seems like it would be potentially preferable to keep the Rayleigh option from LBLRTM on the table as a backup.*

**The v13 predictors were implemented this way because several RTTOV users requested separable molecular Rayleigh extinction. It was also desirable to be able to separate molecular Rayleigh extinction from the gas absorption because it allows full Rayleigh multiple scattering to be included when using the Discrete Ordinates Method solver, which in turn is used to train the MFASIS fast visible cloud parameterisation look-up tables yielding more accurate simulations. It would be straightforward to include the LBLRTM Rayleigh scattering in the training and turn off the internal Rayleigh extinction parameterisation in RTTOV if required. We have not yet determined the approach we will take for Rayleigh scattering in the UV.**

*Also, Rayleigh scattering has a polarization dependence, which does not seem to be accounted for here.*

**As noted below in response to the general comment about polarisation, RTTOV v13 is not a polarised RT model and so we were not considering polarisation during RTTOV v13 development: this is planned for the next major release of RTTOV (v14). With regards to Rayleigh scattering, the v9 predictors did not consider polarisation either, so in that respect there is no change. The v13 and v9 predictors yield very similar results for visible/near-IR channels with Rayleigh scattering included. We have added a final sentence to the end of section 4 (Rayleigh scattering):**

**"Note that RTTOV is currently an unpolarised radiative transfer model and as such polarisation is not taken into account for Rayleigh scattering for the old or new predictors. It is planned to implement fully polarised simulations in a future version."**

*In the work leading up to figure 1, did you recompute the v7, v9, and v13 predictors for $CO_2$ to reflect current values?*

**We use the same RTTOV atmospheric profile training set for all predictors in the paper. The profiles are described in the RTTOV v12 Science and Validation Report (Saunders et al, 2017). The background $CO_2$ profile used in the mixed gases when $CO_2$ is not variable was valid in 2016/2017 (it has a maximum concentration of around 405 ppmv).**

*Figure 1: A clean read of this figure suggests that v7 predictors perform better than v13 on the whole, with notable exception at 13.36 microns.*

**We agree that this figure suggests that the v7 predictors are marginally better than the v13 predictors in many channels, though the differences between the v7/v13 means and standard deviations are small, and all statistics for the v7 and v13 predictors are well within the SEVIRI channel noise. For GEO sensors the v7 predictors are trained for zenith angles up to ~65 degrees, compared to ~85 degrees for the v13 predictors (noted in lines 200-202 of the text). Based on experience we would expect the v7 predictor errors to increase (and indeed to be worse than the v13 predictors) if the training was extended to larger zenith angles, although we didn't try this during the v13 predictor development. If there was a real concern about this, we could also produce v13 predictor coefficients trained over the smaller range of zenith angles (for which we would expect similar or smaller errors), but for practical reasons we want to limit the number of different coefficient files we are generating and providing to users.**

*Across figures 2-8, the additional noise in the visible / near-IR channels suggests that, perhaps, the correction term is not an adequate approach compared to the V7/V9 methods. i suspect that there's an additional correction that would be needed here, but without knowing the specific coding details, it's impossible to speculate what that might be.*

**Figures 2, 3, 7, and 8 do show an increased inter-channel variability in the bias and standard deviation for the v13 predictors for the IASI short-wave IR channels around 2400-2600 cm-1 which corresponds to the tail of the short-wave $CO_2$ band. It's not immediately obvious to us that this**

**indicates a serious issue, although it may suggest the new parameterisation is over-fitting slightly. We will bear this in mind in future evaluations of the new predictors.**

**Figures 4 and 5 are the only ones showing visible/near-IR channels, and it's not clear that the plots show the v13 predictors have more noise. If anything, Fig 4 shows the standard deviations are smaller with the v13 predictors, so it's not clear that there is a problem with the visible/near-IR channels. Happy to discuss further though if we misunderstood.**

*Figure 10,11 really needs a %difference plot to understand the difference in Jacobians, particularly at the upper atmosphere.*

**We considered showing relative difference plots, but the difficulty in cases like this (Jacobians) is that the plots are dominated by extremely large relative differences that occur when the absolute values are small. Where the absolute values are very small, the relative differences are not of practical significance. This was why we chose to show the actual Jacobians: the plots show that the differences are rather small between the v7/8/9 and v13 Jacobians and thus we do not have any particular reason to anticipate any problems in assimilation/retrieval applications.**

*Figure 12: Looks like a software package made these plots, so my recommendation may not be easy to incorporate. It looks like the mean difference and the standard deviation difference are approximately the same order of magnitude, it would be nice to see the mean difference also, similar to the standard deviation difference panel. These could both be on that panel, or on a 4th panel.*

**We have added a fourth panel which plots the difference in absolute bias as suggested. We added the following text within the paragraph describing Figure 12:**

**"Absolute bias differences between the predictor versions are below 0.01 K in all channels with mixed results showing small increases in bias in channels 7–10 and 21 with the v13 predictors, and small reductions in bias in the other channels (Fig. 12d)."**

*General comment: With polarized solvers coming down the pipe, and the importance of polarization to accurate RT calculations, you may want to mention somewhere in the body of this paper about the importance of polarization in transmittance calculations, particularly for Rayleigh scattering.*

**RTTOV v13 is not a polarised RT model and so we did not consider polarisation in this work. We plan to develop a fully polarised version of RTTOV for the next major release and this may well require some modifications to the optical depth prediction (e.g. for Rayleigh scattering as you note). See earlier comment on this topic where we have added a sentence about polarisation.**